



# Ubiquitous Influence of Wildfire Emissions and Secondary Organic Aerosol on Summertime Atmospheric Aerosol in the Forested Great Lakes Region

Matthew J. Gunsch[1], Nathaniel W. May[1], Miao Wen[2], Courtney L. H. Bottenus[2,3], Daniel J. Gardner[1], Timothy M. VanReken[2,†], Steven B. Bertman[4], Philip K. Hopke[5,6], Andrew P. Ault[1,7], Kerri A. Pratt[1,8]

[1]Department of Chemistry, University of Michigan, Ann Arbor, MI
[2]Department of Civil and Environmental Engineering, Washington State University, Pullman, WA
[3]Pacific Northwest National Laboratory, Richland, WA
[4]Department of Chemistry, Western Michigan University, Kalamazoo, MI
[5]Center for Air Resources, Engineering and Science, Clarkson University, Potsdam, NY
[6]Department of Public Health Sciences, University of Rochester School of Medicine and Dentistry, Rochester, NY
[7]Department of Environmental Health Sciences, University of Michigan, Ann Arbor, MI
[8]Department of Earth and Environmental Science, University of Michigan, Ann Arbor, MI
[†]Now at the National Science Foundation, Arlington, VA

*Correspondence to*: Kerri A. Pratt (prattka@umich.edu), Andrew P. Ault (aulta@umich.edu)

***Abstract.*** Long-range aerosol transport affects locations hundreds of kilometers from the point of emission, leading to distant particle sources influencing rural environments that have few major local sources. Source apportionment was conducted using real-time aerosol chemistry measurements made in July 2014 at the forested University of Michigan Biological Station near Pellston, Michigan, a site representative of the remote forested Great Lakes region. Size-resolved chemical composition of individual 0.5 – 2.0 μm particles was measured using an aerosol time-of-flight mass spectrometer (ATOFMS), and non-refractory aerosol mass less than 1 μm ($PM_1$) was measured by a high resolution aerosol mass spectrometer (HR-AMS). The field site was also influenced by air masses transporting Canadian wildfire emissions and urban pollution from Milwaukee and Chicago. During wildfire influenced periods, 0.5 – 2.0 μm particles were primarily aged biomass burning particles (88% by number). These particles were heavily coated with secondary organic aerosol (SOA) formed during



transport, with organics (average O/C ratio of 0.8) contributing 89% of the $PM_1$ mass. During urban-influenced periods, organic carbon, elemental carbon/organic carbon, and aged biomass burning particles were identified, with inorganic secondary species (ammonium, sulfate, and nitrate) contributing 41% of the $PM_1$ mass, indicative of atmospheric processing. With current models under-predicting organic carbon (OC) in this region and biomass burning being the largest combustion contributor to SOA by mass, these results highlight the importance for regional chemical transport models to accurately predict the impact of long-range transported particles on air quality in the upper Midwest United States, particularly considering increasing intensity and frequency of Canadian wildfires.

**1 Introduction**

Atmospheric particulate matter less than 2.5 µm in diameter ($PM_{2.5}$) has significant impacts on air quality, climate, and human health (Calvo et al., 2013; Pöschl and Shiraiwa, 2015). Atmospheric particles directly affect climate by scattering incoming solar radiation and indirectly by acting as cloud condensation (CCN) and ice nuclei (IN) (IPCC, 2013). Increased levels of $PM_{2.5}$ are also linked to increased health risks, particularly respiratory and cardiovascular diseases (Brook et al., 2004; Pope and Dockery, 2006). Particles can impact areas hundreds of kilometers from their sources through long-range transport, with residence times of up to two weeks depending on particle size and chemical composition (Uno et al., 2009). Determining the impact of the long-range transported particles, as well as how they are transformed in the atmosphere during transport, is a critical topic to accurately predict their air quality and climate effects (Ault et al., 2011; Creamean et al., 2013). During transport, particles undergo heterogeneous reactions and gas-particle partitioning, aging the particles and leading primary particles (e.g., soot) to become internally mixed with secondary species, including water, ammonium, nitrate, sulfate, and oxidized organic carbon, thus changing the chemical composition of individual particles (Moffet and Prather, 2009; Riemer and West, 2013). These aging processes are particularly important since chemical composition is directly related to particle properties, including reactivity, hygroscopicity, toxicity, scattering, and absorption properties (Brook et al., 2004; Pöschl, 2005; Calvo et al., 2013; Fierce et al., 2016). Particle properties also differ based on the distribution of chemical species, or mixing state, within a population of particles – whether various chemical species are contained within a single particle



(internally mixed) or within different particles (externally mixed). Particle mixing state representation in models is particularly important (Bauer et al., 2013), especially for predicting aerosol impacts on the climate (Matsui et al., 2013; Fierce et al., 2016).

Long-range transport of atmospheric particles can contribute to both remote and populated locations being out of compliance with air quality regulations (National Research Council and National Academies, 2010). For example, elevated aerosol mass and ozone in Europe, eastern Canada, and northeastern United States has been attributed to transported Canadian wildfire emissions (Forster et al., 2001; Colarco et al., 2004; Müller et al., 2005; Wang et al., 2010b; Dutkiewicz et al., 2011; Miller et al., 2011; Dempsey, 2013; Kang et al., 2014; Dreessen et al., 2016). A multi-day exceedance of the National
Ambient Air Quality Standard for ozone in Maryland during the summer of 2015 was attributed to Canadian wildfire emissions (Dreessen et al., 2016). Similarly, elevated $PM_{2.5}$ observed in New York and Wisconsin has been attributed to Ohio River Valley emissions. Transported pollutants can impact biogenic secondary organic aerosol (SOA) formation in remote locations (Carlton et al., 2010; Emanuelsson et al., 2013; Xu et al., 2015; Rattanavaraha et al., 2016). Finally, prior and on-going studies
through the IMPROVE program in rural locations throughout North America have investigated both transported and local contributions to the aerosol populations (Hand et al., 2011). Uncertainty in the contributions of long-range aerosols and limited measurements in remote areas can lead to inaccuracies in modeling of aerosol source contributions.

Relatively few studies have chemically characterized atmospheric aerosols in the rural Great
Lakes region of the United States (Sheesley et al., 2004; Kim et al., 2005; Kim et al., 2007; Zhang et al., 2009; Jeong et al., 2011; Sjostedt et al., 2011; Kundu and Stone, 2014; Bullard et al., 2017). Except for the major metropolitan areas of Detroit (MI), Chicago (IL), Minneapolis (MN), and Milwaukee (WI), much of the land is characterized by rural agricultural areas and remote forests without significant anthropogenic emissions. A study in the upper peninsula of Michigan conducted by Sheesley et al. (2004)
observed major contributions from secondary organic aerosol from both biogenic and anthropogenic volatile organic compound (VOC) oxidation in the summer. Studies across rural Illinois and Ohio found major atmospheric contributions from secondary sulfate, nitrate, and organic carbon, consistent with aerosol aging during transport (Kim et al., 2005; Kim et al., 2007; Zhang et al., 2009), though these





locations were much less forested than the more northern Great Lakes regions. Kundu and Stone (2014) measured composition and sources at rural locations in Iowa, identifying major PM mass contributions from biomass burning, combustion, and dust. Jeong et al. (2011) and (Sjostedt et al., 2011) identified contributions from secondary organic aerosol, elemental carbon, and dust in rural Harrow, Ontario,

downwind of Detroit and Windsor. The scarcity of measurement data in the rural Great Lakes region provides limited opportunities for model evaluation and requires assumptions of background primary aerosol.

In remote regions, there are challenges in distinguishing and identifying primary and secondary aerosol sources, particularly for bulk methods (Pratt and Prather, 2012). Single-particle mass

spectrometry allows the identification of particle sources through comparisons with source 'fingerprints' and particle aging through characterization of individual particle chemical mixing state (Pratt and Prather, 2009, 2012). Therefore, to apportion the sources of the aerosol population influencing remote northern Michigan, single particle mass spectrometry measurements were conducted during July 2014 at the University of Michigan Biological Station (UMBS) near Pellston, MI. In this study, individual particle

chemical composition, measured in real-time using single-particle mass spectrometry, was used to identify the sources and secondary processing of transported particles at UMBS. In addition, high resolution aerosol mass spectrometry (HR-AMS) measured chemically-resolved mass concentrations of non-refractory aerosol (organics, sulfate, nitrate, ammonium, and chloride) to provide complementary mass-based characterization of the transported particles at UMBS.

## 2 Methods

### 2.1 Field Site and Instrumentation

Atmospheric measurements were conducted from July 13-24, 2014 at the University of Michigan Biological Station (UMBS) near Pellston, MI, a 10,000-acre, remote, forested location with little local pollution (Carroll et al., 2001). The closest major cities are Milwaukee (370 kilometers southwest),

Detroit (385 kilometers south), and Chicago (466 kilometers southwest). Instrumentation was located



within a laboratory at the base of the Program for Research on Oxidants: Photochemistry, Emissions, and Transport (PROPHET) tower, a 30-meter tall sampling tower (45°33'31"N, 84°42'52"W) (Carroll et al., 2001). Air was sampled from 34 m above ground level (~14 m above the forest canopy) through foam-insulated 1.09-cm I.D. copper tubing at a flow rate of 9.25 L min$^{-1}$ (laminar) with a residence time of 15

s. This tubing was connected to a shared sampling manifold at the base of the tower, allowing individual instruments to each have a dedicated sampling line while limiting particle loss.

An aerosol time-of-flight mass spectrometer (ATOFMS model 3800, TSI, Inc., Shoreview, MN) (Gard et al., 1997; Dall'Osto et al., 2004), described briefly below, was used to measure the size and chemical composition of individual atmospheric particles ranging from 0.5 – 2.0 μm in vacuum

aerodynamic diameter ($d_{va}$) (Section 2.2). An Aerodyne high resolution aerosol mass spectrometer (HR-AMS) (DeCarlo et al., 2006) measured chemically-resolved mass concentrations of non-refractory fine particulate material (nominal vacuum aerodynamic diameter range of 0.05 – 1.0 μm) from July 15–24, 2014. Concentrations for major composition classes (organics, sulfate, nitrate, ammonium, and chloride) are reported here. O/C ratios were calculated throughout the study using the methods described by

Canagaratna et al. (2015). The operation of the HR-AMS followed standard practice as described elsewhere (Jayne et al., 2000; Allan et al., 2003; Jimenez et al., 2003; Allan et al., 2004); the sampling resolution for the UMBS observations was 2.5 min. Calibrations for instrument flow rate, particle sizing and transmission, and ionization efficiency were conducted during the study following documented procedures (Jimenez and DeCarlo, 2017). Data were analyzed using SQUIRREL (version 1.60) and the

high resolution analysis software tool PIKA (version 1.20) (Sueper, 2010), with the concentrations corrected based on the estimated composition-dependent collection efficiency (Middlebrook et al., 2012). Additional instrumentation included an ozone analyzer (Thermo Scientific model 49), a scanning mobility particle sizer spectrometer (SMPS, TSI model 3936) with a sheath flow rate of 4 L/min and an aerosol flow rate of 0.4 L/min for measuring size-resolved number concentrations of mobility diameter particles

12-600 nm, and an aerodynamic particle sizer spectrometer (APS, TSI model 3321) for measuring size-resolved number concentrations of 0.5-19 μm aerodynamic diameter particles. SMPS and APS size distributions were merged to give a continuous aerosol distribution from 0.01-2.5 μm (aerodynamic



diameter) using previously established methods (Khlystov et al., 2004), assuming a density of 1.5 g cm$^{-3}$ and shape factor of 1.

## 2.2 Aerosol Time-of-Flight Mass Spectrometer (ATOFMS)

Using the ATOFMS, 11,430 individual atmospheric particles ranging from 0.5 – 2.0 μm in $d_{va}$ were chemically analyzed from July 13–24, 2014. The design and operation of the ATOFMS has been described in detail elsewhere (Dall'Osto et al., 2004; Su et al., 2004). Briefly, particles are focused through an aerodynamic lens system and optically detected by two 532 nm continuous wave lasers spaced 6 cm apart. Particle aerodynamic diameter is determined from particle velocity, which was calibrated using monodispersed spherical polystyrene latex spheres (0.4 – 2.5 μm, Polysciences, Inc.) of known diameter and density. Particles are individually desorbed and ionized by a 266 nm Nd:YAG laser that was operated at ~1.2 mJ, and the resulting ions enter a dual-polarity reflectron time-of-flight mass spectrometer. Positive and negative ion mass spectra corresponding to the same individual particles are collected. Mass spectral peak lists for individual particles were generated using TSI MS-Analyze software.

The individual particle mass spectra were analyzed using YAADA (yaada.org), a software toolkit for MATLAB. Particles were clustered in YAADA using the ART-2a algorithm with a vigilance factor of 0.80 and a learning rate of 0.05 for 20 iterations (Song et al., 1999). The top 50 clusters were manually classified into five particle types described in section 3.1. These top 50 clusters contained 92% of the 11,430 particle mass spectra collected and are the focus of the manuscript. Particle identification was based on characteristic ATOFMS mass spectral signatures previously described (Silva et al., 1999; Pastor et al., 2003; Qin et al., 2012). The errors associated with number fractions for each particle type were calculated using binomial statistics.

To obtain chemically-resolved number and mass concentrations for 0.5-2.0 μm particles, ATOFMS particle counts were scaled with the APS size-resolved particle number concentration data using the method of Qin et al. (2006) to account for size-dependent particle transmission in the inlet. Briefly, ratios of APS number concentration to ATOFMS non-scaled number concentration were calculated every three hours for each individual size bin defined by the APS for use as a scaling factor.



This scaling factor was then multiplied by the corresponding ATOFMS number concentration, providing size and chemically-resolved particle number concentrations for each of the four particle types. These number concentrations were then converted to mass concentrations using assumed spherical shape and compositionally-specific densities. The following densities were applied for the four particle types: 1.5 g cm$^{-3}$ for biomass burning, 1.5 g cm$^{-3}$ for salts, and 1.25 g cm$^{-3}$ for organic carbon-sulfate (OC-sulfate) and elemental carbon/organic carbon – sulfate (ECOC-sulfate) particles (Spencer et al., 2007; Moffet et al., 2008).

# 3 Results and Discussion

## 3.1 Overview

The UMBS campaign (July 13-24, 2014) was characterized by air masses from three primary directions: north, northwest, and southwest (Figures S1 and S2), representative of periods observed during previous UMBS summer studies (Cooper et al., 2001; VanReken et al., 2015). Analysis of NOAA HYSPLIT backward air mass trajectories showed four distinct air mass time periods (Figure S2). From July 13–15, air primarily came from northwestern Canada. From July 15–17, the wind shifted and came from directly north crossing over Lake Superior and Lake Michigan before arriving at the field site. In contrast, from July 17–22 the air came mainly from south-southwest of the field site, crossing over the major metropolitan areas of Chicago and Milwaukee followed by Lake Michigan. Finally, from July 23–24, air came from the north-northwest of the field site, crossing Lake Superior and Lake Michigan from northern Canada (Figure S2). During summer 2009, VanReken et al. (2015) found that 60% of the air masses came from north/northwest of UMBS, similar to this study (57%). Air came from southern polluted regions 43% of the time during our study, compared to 29% during July-August 2009 (VanReken et al., 2015).

Total PM$_{2.5}$ number, PM$_{2.5}$ mass, and ozone concentrations ranged from 143 to 6,031 particles cm$^{-3}$ (average ± standard deviation: 1,822 ± 1,181 particles cm$^{-3}$), 1 to 43 μg/m$^3$ (average ± standard deviation: 8 ± 8 μg/m$^3$) and 9 to 63 ppb (average ± standard deviation: 32 ± 14 ppb), respectively (Figure 1). Maximum concentrations were detected when the air arrived from the southwestern urban areas, and



the minimum values were observed for air masses from the north during remote air transport (Figures S3 and S4). Previously, VanReken et al. (2015) observed an 85% increase in particle number concentration when air originating from these southwestern urban areas impacted UMBS. These results suggest a wide range of sources affecting the field site, which were directly observed by the ATOFMS. Here, we examine the influences of wildfires (Section 3.3) and urban pollution (Section 3.4) on summertime aerosol chemical composition, compared to remote background (Section 3.2), at UMBS.

Major individual particle types observed by ATOFMS included biomass burning, organic carbon-sulfate (OC-sulfate), and elemental carbon/organic carbon-sulfate (ECOC-sulfate) (Figure 2). Biomass burning particles were characterized by intense peaks at $m/z$ 39 ($K^+$) and -97 ($HSO_4^-$), as well as less intense peaks at $m/z$ 12 ($C^+$), 18 ($NH_4^+$), and 27 ($C_2H_3^+$) (Pratt et al., 2010). Biomass burning particles also contained a peak at $m/z$ 43 ($C_2H_3O^+$), a marker for oxidized OC on particles, which is addressed further in section 3.3. Biomass burning was the most prominent particle type, comprising ~80% of submicron (0.5 – 1.0 μm) and ~50% of supermicron (1 – 2 μm) particles, by number, throughout the study, with number fraction varying according to the level of influence from wildfires. OC-sulfate particles contributed ~7%, by number, to submicron (0.5 – 1.0 μm) particles and ~8%, by number, to supermicron (1.0 – 2.0 μm) particles and were characterized by intense peaks at $m/z$ 27 ($C_2H_3^+$), 39 ($C_3H_3^+/K^+$), +/-43 ($C_2H_3O^{+/-}$), and -97 ($HSO_4^-$). OC-sulfate particles can originate from a variety of sources including primary vehicular emissions (Toner et al., 2008) and secondary organic sources (Pratt and Prather, 2009). The intense $m/z$ 43 (most intense OC-sulfate particle ion peak) is indicative of significant SOA coatings on combustion particles, including biomass burning (Pratt and Prather, 2009). ECOC-sulfate particles, characterized by $C_n^+$ fragment peaks, observed at $m/z$ 12 ($C^+$), 24 ($C_2^+$), 36 ($C_3^+$), 48 ($C_4^+$), etc., as well as markers at $m/z$ 27 ($C_2H_3^+$), 18 ($NH_4^+$), and -97 ($HSO_4^-$), are attributed to vehicular emissions (Toner et al., 2006; Toner et al., 2008) and contributed ~5%, by number, to both sub- and supermicron particles with the majority observed on July 22 during an urban-influenced air mass. In addition to the previously mentioned combustion and secondary particles, Na and Ca salts internally mixed with nitrate were episodically detected, primarily during July 16–18 and July 24–25; these salts may have originated from the Great Lakes (Axson et al., 2016) and/or seawater and are the focus of an upcoming manuscript. For each of the discussed particle types, we present the chemical mixing state by





reporting the number percentage of particles within each particle type that contain a mass spectral marker corresponding to each secondary aerosol chemical species of interest, including sulfate ($HSO_4^-$, $m/z$ -97), nitrate ($NO_2^-$, $m/z$ -46, and/or $NO_3^-$, $m/z$ -62), ammonium ($NH_4^+$, $m/z$ 18), and oxidized OC ($C_3H_2O^-$, $m/z$ -43, or $C_3H_2O^+$, $m/z$ 43) (Qin et al., 2012).

5      PM$_1$ mass measured by the HR-AMS was on average 73% organics (7.8 μg/m$^3$) throughout the study, with a substantial contribution from oxidized organics as determined by an average HR-AMS O/C ratio of 0.84 and through the ATOFMS oxidized organic carbon ion marker $m/z$ 43, $C_2H_3O^+$ (Aiken et al., 2008; Qin et al., 2012). O/C ratios between 0.6 – 1 are commonly associated with low volatility oxidized organic aerosol (LV-OOA) that has undergone extensive aging (Jimenez et al., 2009), consistent with the single-particle observation that SOA coated the major particle types. In addition, the ammonium balance of predicted ammonium versus measured ammonium throughout the study (Figure S5) shows a slight deficit in measured ammonium, typically indicative of acidic aerosol or the presence of organic nitrates/sulfates (Farmer et al., 2010). Also consistent with atmospheric processing during long-range transport, 92% of all 0.5 – 2.0 μm particles, by number, were measured by the ATOFMS to be internally mixed with secondary species, including sulfate ($HSO_4^-$, $m/z$ -97), nitrate ($NO_2^-$, $m/z$ -46 and/or $NO_3^-$, $m/z$ -62), ammonium ($NH_4^+$, $m/z$ 18), and/or oxidized OC ($C_3H_2O^-$, $m/z$ -43 or $C_3H_2O^+$, $m/z$ 43) (Qin et al., 2012). On average, sulfate comprised 20% (2.2 μg/m$^3$) of the total PM$_1$ mass measured by HR-AMS.

## 3.2 Remote Background Air Mass Influence

From July 15-17, air arrived at UMBS originating from rural northern Canada. The average PM$_{2.5}$ number concentration was $903 \pm 499$ particles cm$^{-3}$ (range of 143 - 2163 particles cm$^{-3}$, Figure 1) and average PM$_{2.5}$ mass concentration was $1.9 \pm 0.4$ μg/m$^3$ with a particle number mode of 82 nm (Figure 1 and S3). The average ozone concentration was $17 \pm 6$ ppb (Figure 1). With a lack of direct wildfire influence (Figure 4), $61 \pm 1\%$ of the 0.5 – 2.0 μm particles, by number, were classified by ATOFMS as aged biomass burning aerosols, relatively similar to the background biomass burning particle influence reported by Hudson et al. (2004) and Pratt et al. (2010) for the United States free troposphere (33-52% by number). Biomass burning particles were internally mixed with oxidized OC ($80 \pm 2\%$, by number) or




mixed with sulfate ($85 \pm 2\%$). Nitrate was internally mixed with $8 \pm 2\%$, by number of biomass burning particles and $33 \pm 3\%$, by number, of OC-sulfate particles. It is likely that, while the observed biomass burning particles have a small potassium-rich (biomass burning) core, they are primarily SOA by mass (Pratt and Prather, 2009; Moffet et al., 2010) (Section 3.3). The HR-AMS showed average $PM_1$ organic

mass concentrations of 4.4 μg/m$^3$, with minimal contribution from sulfate (0.3 μg/m$^3$), as well as nitrate and ammonium (both less than 0.1 μg/m$^3$ on average) (Figure 3).

The significant internal mixing of oxidized OC combined with the significant organic mass loading (average HR-AMS O/C ratio of 0.9) is consistent with high SOA mass on the particles (Aiken et al., 2008). Previous studies in rural and forested environments found similarly high O/C ratios during

periods of non-polluted air and attributed this to regional SOA formation (Jimenez et al., 2009; Sun et al., 2009; Raatikainen et al., 2010; Sjostedt et al., 2011). There was a notable spike in O/C ratio on July 15 – 16 to 1.2, indicative very highly oxidized organics. O/C ratios of this magnitude have previously been observed at the remote Whistler Mountain, where organic aerosol O/C ratios up to ~1.3 where observed during organic aerosol accumulation events (Sun et al., 2009). Sheesley et al. (2004) found that SOA,

primarily biogenic-derived, contributed over 90% of the total organic carbon mass observed during the summer at the Seney National Wildlife Refuge in northern Michigan, located 120 km northwest of UMBS. Notably, ultrafine particle growth was observed at UMBS on July 16 during this high O/C ratio spike (Gunsch et al., 2017). The air arriving during this period was not under the influence of wildfires (Section 3.3) or urban areas (Section 3.4), and is therefore expected to be representative of remote

background conditions.

### 3.3 Wildfire Influence

From July 13-15 and July 24 mid-day through July 25, the NOAA Hazard Mapping System (HMS) Smoke Product (Rolph et al., 2009) indicated that smoke plumes originating from wildfires within the Northwest Territories (Canada) directly influenced UMBS (Figure 4). According to the Canadian

Interagency Forest Fire Centre, over 5,500 km$^2$ of land burned within the Northwest Territories during July 2014 (CIFFC, 2014). Canadian wildfires are a major source of global $PM_{2.5}$, with estimates of ~1.6





Tg yr$^{-1}$ emitted to the atmosphere (Wiedinmyer et al., 2006). Average PM$_{2.5}$ number and concentrations during these two wildfire influence periods were statistically higher (t-test, α = 0.05) at 1400 ± 800 particles cm$^{-3}$ (range of 147 – 4832 particles cm$^{-3}$, Figure 1) and 5.4 ± 2.6 μg/m$^3$ (range of 1.3 – 10.5 μg/m$^3$, Figure 1), respectively, compared to the background period (Section 3.2). The particle mode during wildfire influence was 80 nm, similar to background periods (Figure S3). Ozone was also elevated during July 13-15 reaching as high as 35 ppb, compared to an average of 10 ppb during the background period (Figure 1). During these periods, the air masses did not pass over any major urban areas (Figure S2), making ozone production within the smoke plume during transport the likely source (Jaffe and Wigder, 2012). Ozone did not increase during the July 24 smoke plume, staying near the average for the study (25 ± 12 ppb) with a concentration of 26 ± 3 ppb (Figure 1). While an ozone increase is often observed for aged wildfire plumes, an increase does not always occur during wildfire influence, such as when low NO$_x$ levels within plumes, potentially due to smoldering combustion, limit the production of ozone (Jaffe and Wigder, 2012).

During the wildfire influenced periods, 88 ± 1% of the measured 0.5 – 2.0 μm particles, by number, were biomass burning particles, with an average mass concentration of 0.42 μg/m$^3$ (Figure 5) and a maximum of 0.80 μg/m$^3$ occurring during the early afternoon of July 14 when the heaviest wildfire smoke was reported by the NOAA smoke product (Figure 4A). Minor contributions of OC-sulfate particles (8 ± 1% by number) were also measured. The OC-sulfate particle mass spectra (Figure 2B) showed that 75 ± 5%, by number, contained potassium (K$^+$, $m/z$ 39), suggesting that these were highly aged biomass burning particles coated by SOA such that the typical biomass burning mass spectral signature had been masked, as observed previously by Pratt and Prather (2009) using a thermodenuder. These OC-sulfate particles featured a dominant intense $m/z$ 43 (C$_2$H$_3$O$^+$) ion peak, indicating that these particles were heavily coated with SOA. During the afternoon event on July 24, PM$_1$ organic mass concentrations measured by HR-AMS nearly doubled from 2.5 ± 0.1 μg/m$^3$ before the event to 4.5 ± 0.3 μg/m$^3$ during the event (Figure 3), accounting for ~90% of the total PM$_1$ mass concentration. The HR-AMS O/C ratio was 0.8 during wildfire periods, consistent with biomass burning particles heavily coated with SOA (Aiken et al., 2008), as also observed by 95 ± 1%, by number, of the biomass burning and OC-



sulfate particles, measured by ATOFMS during these periods, featuring the oxidized OC ion marker ($m/z$ 43, $C_2H_3O^+$) (Figure 6). Freshly emitted biomass burning has a O/C ratio of ~0.2, which can increase to ~0.6 in only a few hours as oxidized material condenses onto the particles (Grieshop et al., 2009; Pratt et al., 2011). As the wildfire air masses measured during the present study were transported over multiple

days over Canadian forests, biogenic SOA, from condensation of monoterpene oxidation products (Slowik et al., 2010), likely contributed to the observed O/C ratio of 0.8 at UMBS.

During transport of the biomass burning aerosols, accumulation of sulfate also occurred, with 97 ± 1%, by number, of biomass burning particles internally mixed with sulfate ($m/z$ -97, $HSO_4^-$) (Figure 6). HR-AMS measured $PM_1$ sulfate also increased from less than 0.1 μg/m³ to 2 μg/m³ after mid-day July 24

(Figure 3). Increases in particulate sulfate mass have been observed during wildfire plume aging (DeBell et al., 2004; Pratt et al., 2010). In comparison, the HR-AMS measured limited amounts of $PM_1$ ammonium (~2% of total mass, 0.2 μg/m³) during the wildfire event on July 24 (Figure 3). However, ammonium was internally mixed in 38 ± 2%, by number, of biomass burning and 68 ± 2%, by number, of OC-sulfate particles (Figure 6). This result indicates that while ammonium was present within many particles, it was

a minor fraction of the particle mass. Nitrate was also internally mixed with 43 ± 2% of biomass burning particles, by number, and 17 ± 2%, by number, of OC-sulfate particles (Figure 6), and the HR-AMS only measured ~1% of $PM_1$ mass to be nitrate (0.06 μg/m³). Therefore, it is likely that the ammonium was present in the form of ammonium sulfate internally mixed with biomass burning and OC-sulfate.

**3.4 Urban Air Mass Influence**

From July 17-22, UMBS was influenced by air masses from the southwest, passing over the major metropolitan areas of Chicago and Milwaukee before arriving at the site (Figure S2) after transport times of 24-36 hours. The average ozone concentration was elevated at an average of 41 ± 12 ppb similar to previous measurements by Cooper et al. (2001) at UMBS when under the direct influence of urban pollution (Figure 1). The $PM_{2.5}$ number and mass concentration for this period were 2,700 ± 900 particles

cm⁻³ (range of 414 – 6,031 particles cm⁻³) and 14 ± 8 μg/m³ (range of 2 – 43 μg/m³), respectively, the highest for the study (Figure 1). The particle mode of 69 nm was also the smallest of the study (Figure



S3). VanReken et al. (2015) previously observed the highest particle number concentrations (3,000 ± 1,300 particles cm$^{-3}$) at UMBS during the influence of southern air masses. Wildfire smoke influence was present during this period as shown by the NOAA smoke product (Figure 4C). However, unlike during the previous periods, this smoke originated mainly from the southern United States (active fires were located in Tennessee, Arkansas, and Missouri). Biomass burning particles measured by ATOFMS steadily increased in mass concentration throughout this period (Figure 5), with a notable spike in the mass concentration on July 22 observed in both the submicron (0.5-1.0 µm particles: 2.3 µg/m$^3$) and supermicron (0.3 µg/m$^3$) size ranges (1.0-2.0 µm particles: Figure 5). Overall, during urban influence, biomass burning particles accounted for 72 ± 2% of the particles, by number, and ~30% of the total mass concentration (Figure 5). The biomass burning particles were aged, as shown by internal mixtures of sulfate (88 ± 2%, by number), oxidized OC (92 ± 1%, by number), ammonium (58 ± 2%, by number), and nitrate (30 ± 2%, by number) (Figure 6). The greatest internal mixing with ammonium was observed during this period. The HR-AMS also measured the highest average ammonium mass concentration during this period of 1.6 µg/m$^3$, accounting for 10% of the total PM$_1$ particle mass (Figure 3). Agricultural activities, both crop and livestock, located to the south and southwest of the field site (Stephen and Aneja, 2008; Paulot et al., 2014) may be the source of the elevated ammonium levels.

ECOC-sulfate and OC-sulfate particles comprised the second most prominent particle types measured by ATOFMS during this urban-influenced period at 12 ± 1% and 9 ± 1% of the submicron (0.5 – 1.0 µm) particles, by number, and an average of 0.08 µg/m$^3$ and 0.03 µg/m$^3$, respectively (Figure 5). The influence of urban vehicular combustion resulted in the increased levels of measured ECOC-sulfate particles (Toner et al., 2006; Toner et al., 2008), compared to non-urban influenced periods (2 ± 1% by number). HR-AMS PM$_1$ mass concentrations (Figure 3) showed increased organic mass during urban influence with an average mass concentration of 9.7 µg/m$^3$ (Figure 3), likely due to a mixture of biomass burning, anthropogenic, and biogenic organic aerosol. The average HR-AMS O/C ratio during the urban period was the lowest of the study (0.78), likely due to increased contributions from hydrocarbon-like organic aerosol from urban vehicle combustion emissions (Aiken et al., 2008), in contrast to primarily oxidized organic aerosol during regional background periods (Jimenez et al., 2009). An increase in less





oxidized organic aerosol was similarly observed in rural Ontario when the site was influenced by urban air masses from Detroit, compared to remote air masses (Sjostedt et al., 2011). The ECOC-sulfate and OC-sulfate particles were highly aged, with ~75%, by number, of each particle type internally mixed with ammonium, consistent with particle aging during transport (Figure 6C). Ammonium (1.6 μg/m$^3$) and sulfate (4.9 μg/m$^3$) comprised over 40% of the total PM$_1$ mass measured by the HR-AMS during these periods, likely in the form of ammonium sulfate (Figure 3). Urban influenced air masses had the highest mass concentration of sulfate (up to 10 μg/m$^3$) measured throughout the study. In contrast, there was little presence of nitrate internally mixed in the ECOC-sulfate (4 ± 2%, by number) and OC-sulfate (19 ± 5%, by number) particles (Figure 6), and nitrate only comprised 1% (0.2 μg/m$^3$) of the total PM$_1$ mass concentration from the urban influence (Figure 3).

## 4 Conclusions

Source apportionment of atmospheric particles in the summertime was conducted at the forested University of Michigan Biological Station, located in remote northern Michigan. The field site was impacted by air masses from three distinct areas: remote background, northwestern Canada, and southwestern urban areas. July 2014 was one of the most active burning seasons for the Northwest Territories in over two decades with a total of 10,643 km$^2$ of land burned, significantly more than the ten-year (1,944 km$^2$) and twenty-five year (2,423 km$^2$) averages (CIFFC, 2014). The increased wildfire activity noticeably impacted northern Michigan, as the presence of biomass burning particles was ubiquitous throughout the study and made up the majority of measured particle number and mass concentrations. While air also came from urban areas southwest of UMBS, aged biomass burning particles dominated particle number concentrations due to wildfire influences from the southern United States. Due to the urban influence, these air masses had the highest mass contributions of sulfate (over 50 times the background) detected during the entire study. The accumulation of soluble secondary species, including sulfate and nitrate, increases the CCN ability of biomass burning particles (Furutani et al., 2008; Petters et al., 2009; Wang et al., 2010a), illustrating the importance of transported wildfire emissions.





While biomass burning particles were the most dominant particle core detected, SOA was a major contributor to particle mass during the study. On average, the HR-AMS organic aerosol O/C ratio was 0.84, indicative of highly oxidized organic carbon (Aiken et al., 2008). During remote background periods, internal mixing of oxidized OC combined with the significant $PM_1$ organic mass loading is indicative of the high mass loading of biogenic SOA in the forested region (Sheesley et al., 2004). During wildfire-influenced air masses, organics contributed ~90% to the $PM_1$ mass, with SOA internally mixed with biomass burning and OC-sulfate particles, indicating that SOA from both biogenic VOC oxidation and wildfire combustion is a major source of OC in the region. Models under-predict OC in this region, and Jathar et al. (2014) indicates that on a national level, models predict biomass burning is the largest combustion contributor to SOA by mass, consistent with the significant influence of wildfires during this work.

Modeling studies have called for further investigations of wildfire emissions and areas they affect in order to reduce uncertainty within models due to limited data, particularly when modeling interactions between wildfire plumes and urban emissions. Wildfire plume ozone production can lead to areas far from the original source to be out of compliance with regulatory standards, demonstrating the importance to be able to accurately model ozone production (Hu et al., 2008; Jaffe and Wigder, 2012; Lu et al., 2016). Also, as described here, particles aged through transport show internal mixtures of nitrate, sulfate and oxidized organics, which can lead to increased CCN activity (Furutani et al., 2008). With wildfires expected to increase in both intensity and frequency due to climate change (Gillett et al., 2004; Liu et al., 2010; Knorr et al., 2016; Veira et al., 2016), the contributions of long-range transported biomass burning emissions to the upper Midwest US atmosphere are expected to increase, such that air quality modeling efforts will need to supplement their existing emissions to account for the expected increase in wildfire emissions (Smith and Mueller, 2010).

*Competing Interests.* The authors declare that they have no competing financial interests.

*Acknowledgements.* Funding for the UMBS study was provided by University of Michigan MCubed Program and UMBS graduate fellowships for M. Gunsch, N. May, and D. Gardner. We also thank the





Pratt and Ault Groups for assistance on the field study. Support for T. VanReken and M. Wen was provided by the U.S. Department of Energy Early Career Research Program (award no. SC0003899). Jennifer Dean (Washington State University) is thanked for her assistance during the field study. Donna Sueper (Aerodyne) is thanked for assistance with HR-AMS data processing. The authors gratefully

5   acknowledge the NOAA Air Resources Laboratory (ARL) for the provision of the HYSPLIT transport and dispersion model and READY website (http://www.ready.noaa.gov) used in this publication. The authors also gratefully acknowledge the NOAA Office of Satellite and Product Operations for the use of the Hazard Mapping System Smoke Product (http://www.ospo.noaa.gov/Products/land/hms.html).



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





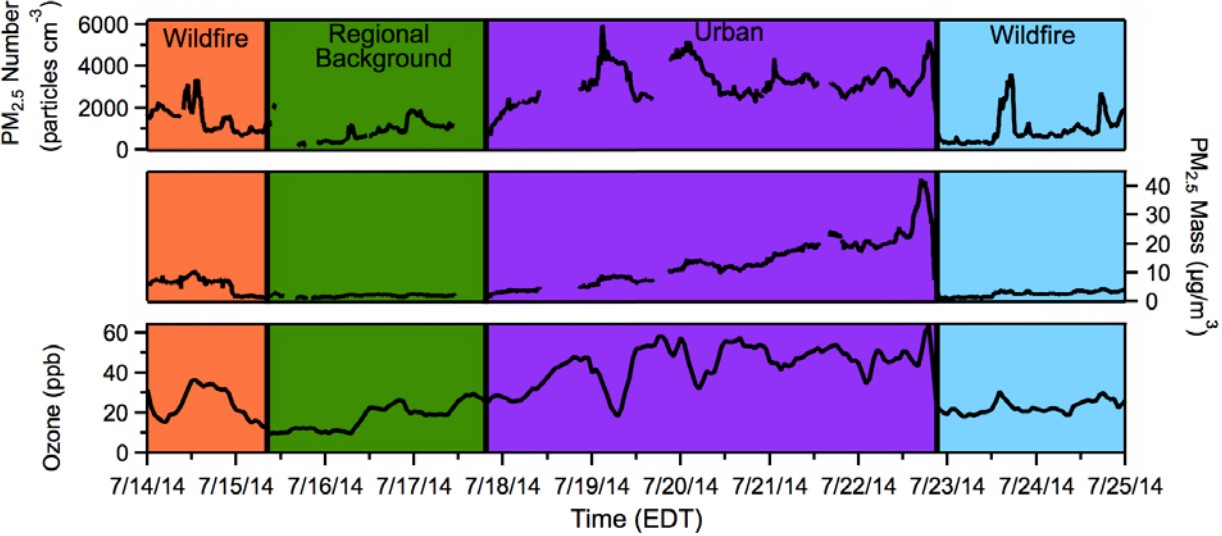

**Figure 1.** Time-resolved PM$_{2.5}$ number and mass concentrations and ozone mole ratios during the different periods of air mass influence. Periods without data are due to instrument down time. Colors of the different time periods correspond to the colors of the corresponding HYSPLIT backward air mass trajectories in Figure S2.





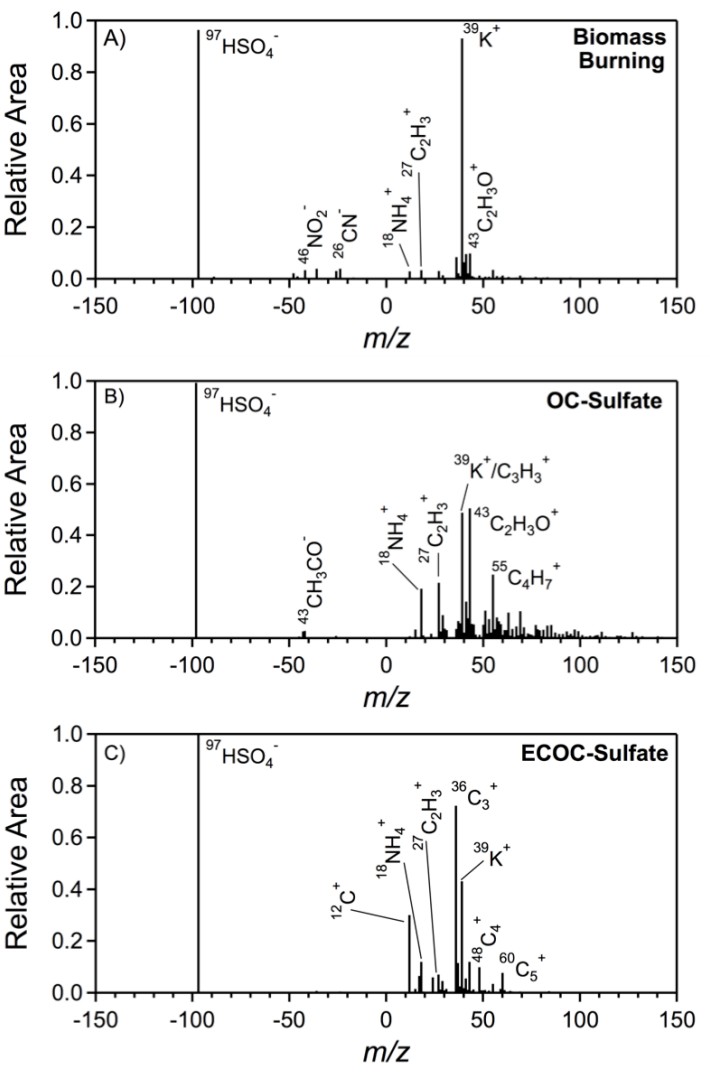

**Figure 2.** Average positive and negative ion single-particle mass spectra (ATOFMS), with characteristic peaks labeled, for the dominant aged combustion particle types observed: (A) biomass burning, (B) OC-sulfate, and (C) ECOC-sulfate.




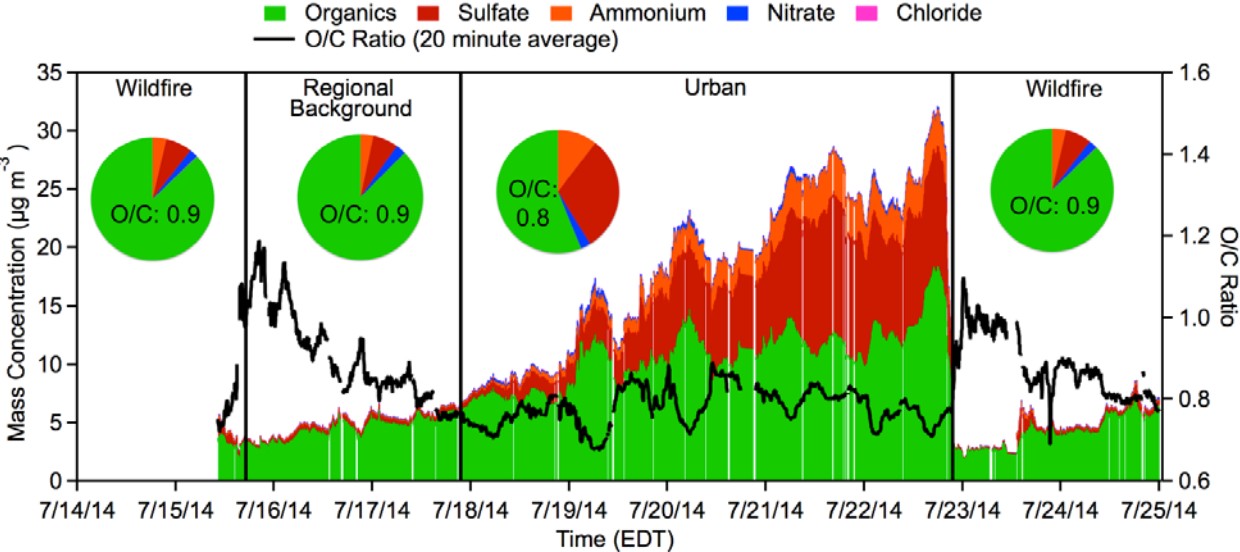

**Figure 3.** PM$_1$ non-refractory chemically speciated mass concentrations, as well as O/C ratios (20 min averages), measured by HR-AMS. Periods of influence are notated and separated by solid vertical lines. Pie charts represent the average mass fractions for each air mass period, with average O/C ratio inset.





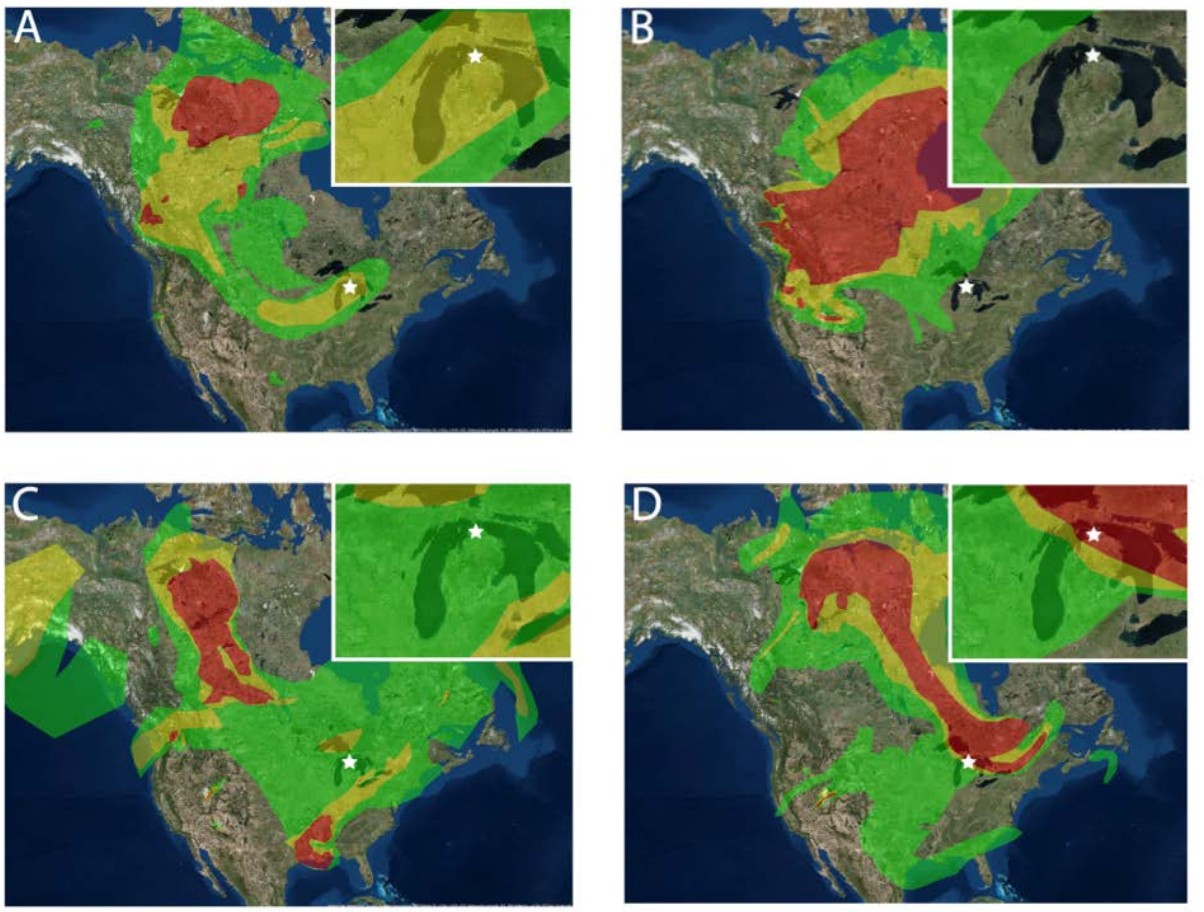

**Figure 4.** Representative NOAA HMS smoke maps for four representative days during the time periods of different air influence: (A) July 14, wildfire influence; (B) July 16, remote background influence; (C) July 21, Urban influence; (D) July 24, wildfire influence. Inset enlarges the state of Michigan to clearly display smoke influence on the field site, shown as a star.



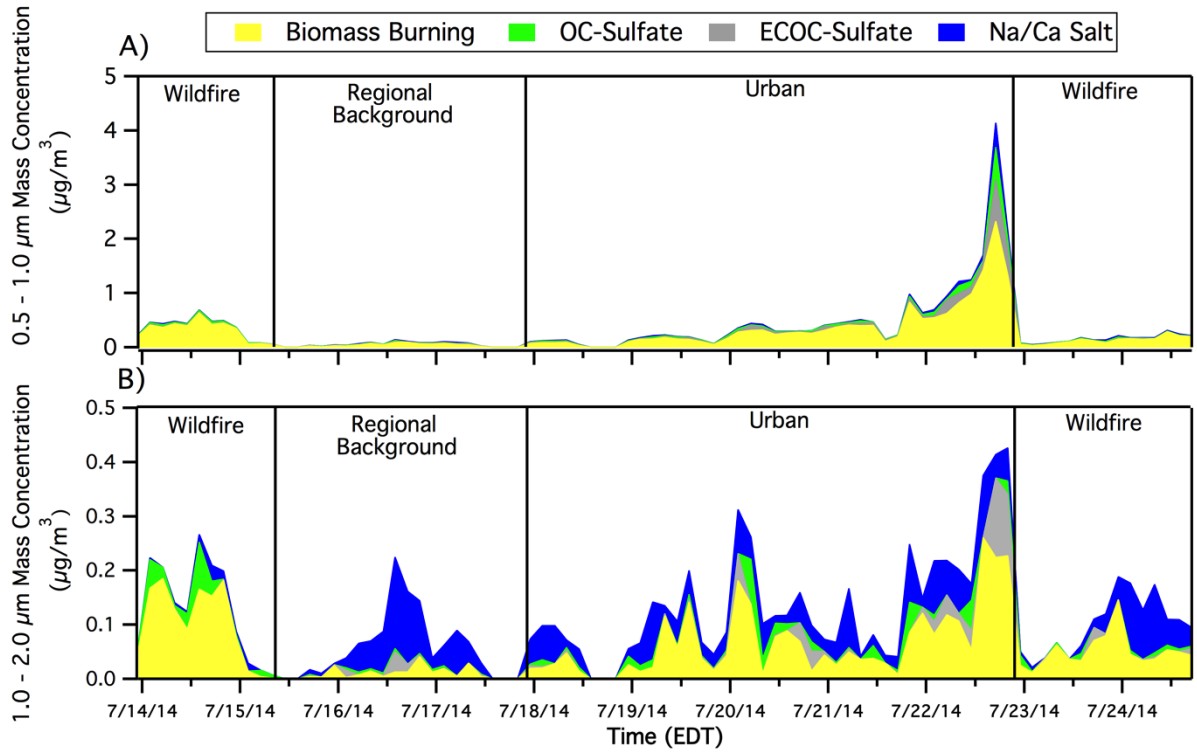

**Figure 5.** Three hour binned mass concentrations of (A) 0.5 – 1.0 µm and (B) 1.0 – 2.0 µm particle types, as measured by ATOFMS. Gaps in the data correspond to periods when APS data were not available for scaling.





**Figure 6.** Number fractions of individual biomass burning, OC-sulfate, and ECOC-sulfate particles that were internally mixed with secondary species, as determined by ATOFMS ion markers, including oxidized OC ($C_2H_3O^+$, *m/z* 43), ammonium ($NH_4^+$, *m/z* 18), nitrate ($NO_2^-$, *m/z* -46, and/or $NO_3^-$, m/z -62), and sulfate ($HSO_4^-$, *m/z* -97). Since a given particle can contain more than one secondary species, the number fractions can add to greater than one for a given particle type. Chemical mixing states are provided here for the four air mass time periods: (A) wildfire influence from July 13-15, (B) clean air from northern Canada from July 15-17, (C) mix of wildfire and urban influences from July 17-22, (D) mix of clean air and Canadian wildfires from July 23-24.