# Peer review of "Ubiguitous Influence of Wildfire Emissions and Secondary Organic Aerosol on Summertime Atmospheric Aerosol in the Forested Great Lakes Region"

_Atmospheric Chemistry and Physics, 2017_

## Referee Comment (RC1) · Anonymous Referee #2 · 24 Nov 2017

Gunsch et al. present observations of aerosol concentration and composition at the University of Michigan Biological Station (UMBS) for July 2014. The authors use a combination of a high-resolution time-of-flight aerosol mass spectrometer, single particle aerosol time-of-flight mass spectrometer, and combination scanning mobility particle sizer spectrometer and aerodynamic particle size spectrometer to investigate the aerosol characteristics between 0.01 – 2.5 µm during this time period. The authors found four different air masses impacted the area during the time period: 2 air masses impacted by wildfires from Canada, 1 air mass impacted by cities south of UMBS, and 1 air mass from clean regions over Canada. The authors found an increase in particle number and mass, over the clean regime, for the air masses impacted by wildfires and cities; however, no matter where the air came from, it was always influenced by biomass burning. The paper provides important information about what influences the aerosol mass in a background, rural location, and is of value for *Atmospheric Chemistry and Physics* community; however, there are some concerns and some clarifications that need to be addressed first prior to publication.

**Major Comments**

1) I'm wondering why the AMS data was not used more to help further validate the results from the single particle mass spectrometer, or to support some of the authors' hypotheses. For example, either PMF (Ulbrich et al., 2009), "poor-man's PMF" (Aiken et al., 2009; Zhang et al., 2005), or triangle plots of different fragments (Cubison et al., 2011; Hu et al., 2015; Ng et al., 2010) would further support the evidence of OA being strongly influenced by biomass burning, anthropogenic emissions, and biogenic emissions/chemistry (e.g., page 10, Lines 3 – 5 and page 11, lines 20 – 22). Without this support, speculations that biogenic emissions and chemistry leading to the very high O/C ratios observed is hard to

interpret (page 11, lines 20 – 22). Other studies found high O/C ratios due to photochemical aging of the biomass burning emissions and aerosol (Liu et al., 2016; Zhou et al., 2017), which may have led to the high O/C instead of OA production from biogenic emissions. Also, cloud processing of the gases and OA may lead to high O/C ratios, along with the $SO_4$ (Sullivan et al., 2016). A discussion either including these various processes or a discussion that argues why one process over the others leads to the high O/C ratios is needed.

2) Throughout the Sections 3.2 – 3.4, in the comparisons of the different aerosol regimes, the aerosol number mode is mentioned. The biomass burning mode is the same as the background mode, and the urban mode is smaller than both the biomass burning and background mode; however, the authors discuss how chemistry and accumulation are occurring during transport of the biomass burning and urban air masses. A discussion about why the modes are similar (or smaller) while chemistry and accumulation is occurring is necessary for the readers to better relate these two possibly contradictory processes.

3) A more in-depth analysis of some of the aerosol characteristics would improve the results and the paper. For example, page 12, lines 6 – 11, the authors very briefly discuss an accumulation of $SO_4$ into the biomass burning particles; however, these particles are coming from wildfires in Canada. A discussion about where this $SO_4$ comes from would be beneficial, as recent studies have found a minor role for biomass burning emissions $SO_4$ and unclear for $SO_2$ emissions (Collier et al., 2016; Liu et al., 2016, 2017). As another example, the authors found little $NO_3$ in the air masses impacted by urban regions (page 14, lines 7 – 10 and Figure 3), which is surprising with the $NO_x$ emissions and chemistry.

What would have led to the extremely low amounts, especially since other downwind sites have observed enhanced $NO_3$ (Jimenez et al., 2009)?

**Minor Comments**

1) Page 1, line 27: "The field site was also influenced . . .". It is not clear in the abstract what the influences are before this line.

2) Page 2, line 21: Change leading primary particles to leading to primary particles

3) Page 2, line 22: Why is water included as a secondary species for aerosol?

4) Page 4, line 3: Any reason why Slowik et al. (2011) is not included in this comparison of SOA in Ontario?

5) Page 5, line 19: Please check the Jimenez and DeCarlo, 2017 citation. The website listed for this citation leads to Canadian Interagency Forest Fire Centre.

6) Page 11, line 1: superscript the minus sign

7) Page 11, line 12: Please change limit to limiting

**References**

Aiken, a. C., Salcedo, D., Cubison, M. J., Huffman, J. a., DeCarlo, P. F., Ulbrich, I. M., Docherty, K. S., Sueper, D., Kimmel, J. R., Worsnop, D. R., Trimborn, a., Northway, M., Stone, E. a., Schauer, J. J., Volkamer, R., Fortner, E., de Foy, B., Wang, J., Laskin, a., Shutthanandan, V., Zheng, J., Zhang, R., Gaffney, J., Marley, N. a., Paredes-Miranda, G., Arnott, W. P., Molina, L. T., Sosa, G. and Jimenez, J. L.: Mexico City aerosol analysis during MILAGRO using high resolution aerosol mass spectrometry at the urban supersite (T0) – Part 1: Fine particle composition and organic source apportionment, Atmos. Chem. Phys. Discuss., 9(2), 8377–8427, doi:10.5194/acpd-9-8377-2009, 2009.

Collier, S., Zhou, S., Onasch, T. B., Jaffe, D. A., Kleinman, L., Sedlacek, A. J., Briggs, N. L., Hee, J., Fortner, E., Shilling, J. E., Worsnop, D., Yokelson, R. J., Parworth, C., Ge, X., Xu, J., Butterfield, Z., Chand, D., Dubey, M. K., Pekour, M. S., Springston, S. and Zhang, Q.: Regional Influence of Aerosol Emissions from Wildfires Driven by Combustion Efficiency: Insights from the BBOP Campaign, Environ. Sci. Technol., 50(16), 8613–8622, doi:10.1021/acs.est.6b01617, 2016.

Cubison, M. J., Ortega, A. M., Hayes, P. L., Farmer, D. K., Day, D., Lechner, M. J., Brune, W. H., Apel, E., Diskin, G. S., Fisher, J. A., Fuelberg, H. E., Hecobian, A., Knapp, D. J., Mikoviny, T., Riemer, D., Sachse, G. W., Sessions, W., Weber, R. J., Weinheimer, A. J., Wisthaler, A. and Jimenez, J. L.: Effects of aging on organic aerosol from open biomass burning smoke in aircraft and laboratory studies, Atmos. Chem. Phys., 11(23), 12049–12064, doi:doi:10.5194/acp-11-12049-2011, 2011.

Hu, W. W., Campuzano-Jost, P., Palm, B. B., Day, D. A., Ortega, A. M., Hayes, P. L., Krechmer, J. E., Chen, Q., Kuwata, M., Liu, Y. J., de Sá, S. S., McKinney, K., Martin, S. T., Hu, M., Budisulistiorini, S. H., Riva, M., Surratt, J. D., St. Clair, J. M., Isaacman-Van Wertz, G., Yee, L. D., Goldstein, A. H., Carbone, S., Brito, J., Artaxo, P., de Gouw, J. A., Koss, A., Wisthaler, A., Mikoviny, T., Karl, T., Kaser, L., Jud, W., Hansel, A., Docherty, K. S., Alexander, M. L., Robinson, N. H., Coe, H., Allan, J. D., Canagaratna, M. R., Paulot, F. and Jimenez, J. L.: Characterization of a real-time tracer for isoprene epoxydiols-derived secondary organic aerosol (IEPOX-SOA) from aerosol mass spectrometer measurements, Atmos. Chem. Phys., 15(20), 11807–11833, doi:10.5194/acp-15-11807-2015, 2015.

Jimenez, J. L., Canagaratna, M. R., Donahue, N. M., Prevot, A. S. H., Zhang, Q., Kroll, J. H., DeCarlo, P. F., Allan, J. D., Coe, H., Ng, N. L., Aiken, A. C., Docherty, K. S., Ulbrich, I. M., Grieshop, A. P., Robinson, A. L., Duplissy, J., Smith, J. D., Wilson, K. R., Lanz, V. A., Hueglin, C., Sun, Y. L., Tian, J., Laaksonen, A., Raatikainen, T., Rautiainen, J., Vaattovaara, P., Ehn, M., Kulmala, M., Tomlinson, J. M., Collins, D. R., Cubison, M. J., Dunlea, E. J., Huffman, J. A., Onasch, T. B., Alfarra, M. R., Williams, P. I., Bower, K., Kondo, Y., Schneider, J., Drewnick, F., Borrmann, S., Weimer, S., Demerjian, K., Salcedo, D., Cottrell, L., Griffin, R., Takami, A., Miyoshi, T., Hatakeyama, S., Shimono, A., Sun, J. Y., Zhang, Y. M., Dzepina, K., Kimmel, J. R., Sueper, D., Jayne, J. T., Herndon, S. C., Trimborn, A. M., Williams, L. R., Wood, E. C., Middlebrook, A. M., Kolb, C. E., Baltensperger, U., Worsnop, D. R. and Worsnop, D. R.: Evolution of organic aerosols in the atmosphere., Science, 326(5959), 1525–1529, doi:10.1126/science.1180353, 2009.

Liu, X., Zhang, Y., Huey, L. G., Yokelson, R. J., Wang, Y., Jimenez, J. L., Campuzano-Jost, P., Beyersdorf, A. J., Blake, D. R., Choi, Y., St. Clair, J. M., Crounse, J. D., Day, D. A., Diskin, G. S., Fried, A., Hall, S. R., Hanisco, T. F., King, L. E., Meinardi, S., Mikoviny, T., Palm, B. B.,

Peischl, J., Perring, A. E., Pollack, I. B., Ryerson, T. B., Sachse, G., Schwarz, J. P., Simpson, I. J., Tanner, D. J., Thornhill, K. L., Ullmann, K., Weber, R. J., Wennberg, P. O., Wisthaler, A., Wolfe, G. M. and Ziemba, L. D.: Agricultural fires in the southeastern U.S. during SEAC [4] RS: Emissions of trace gases and particles and evolution of ozone, reactive nitrogen, and organic aerosol, J. Geophys. Res. Atmos., 121(12), 7383–7414, doi:10.1002/2016JD025040, 2016.

Liu, X., Huey, L. G., Yokelson, R. J., Selimovic, V., Simpson, I. J., Müller, M., Jimenez, J. L., Campuzano-Jost, P., Beyersdorf, A. J., Blake, D. R., Butterfield, Z., Choi, Y., Crounse, J. D., Day, D. A., Diskin, G. S., Dubey, M. K., Fortner, E., Hanisco, T. F., Hu, W., King, L. E., Kleinman, L., Meinardi, S., Mikoviny, T., Onasch, T. B., Palm, B. B., Peischl, J., Pollack, I. B., Ryerson, T. B., Sachse, G. W., Sedlacek, A. J., Shilling, J. E., Springston, S., St. Clair, J. M., Tanner, D. J., Teng, A. P., Wennberg, P. O., Wisthaler, A. and Wolfe, G. M.: Airborne measurements of western U.S. wildfire emissions: Comparison with prescribed burning and air quality implications, J. Geophys. Res. Atmos., 122(11), 6108–6129, doi:10.1002/2016JD026315, 2017.

Ng, N. L., Canagaratna, M. R., Zhang, Q., Jimenez, J. L., Tian, J., Ulbrich, I. M., Kroll, J. H., Docherty, K. S., Chhabra, P. S., Bahreini, R., Murphy, S. M., Seinfeld, J. H., Hildebrandt, L., Donahue, N. M., Decarlo, P. F., Lanz, V. A., Prévôt, A. S. H., Dinar, E., Rudich, Y. and Worsnop, D. R.: Organic aerosol components observed in Northern Hemispheric datasets from Aerosol Mass Spectrometry, Atmos. Chem. Phys., 10(10), 4625–4641, doi:10.5194/acp-10-4625-2010, 2010.

Slowik, J. G., Brook, J., Y.-W. Chang, R., Evans, G. J., Hayden, K., Jeong, C. H., LiS.-M., Liggio, J., Liu, P. S. K., McGuire, M., Mihele, C., Sjostedt, S., Vlasenko, A. and Abbatt, J. P. D.: Photochemical processing of organic aerosol at nearby continental sites: Contrast between urban plumes and regional aerosol, Atmos. Chem. Phys., 11(6), 2991–3006, doi:10.5194/acp-11-2991-2011, 2011.

Sullivan, A. P., Hodas, N., Turpin, B. J., Skog, K., Keutsch, F. N., Gilardoni, S., Paglione, M., Rinaldi, M., Decesari, S., Facchini, M. C., Poulain, L., Herrmann, H., Wiedensohler, A., Nemitz, E., Twigg, M. M. and Collett Jr., J. L.: Evidence for ambient dark aqueous SOA formation in the Po Valley, Italy, Atmos. Chem. Phys., 16(13), 8095–8108, doi:10.5194/acp-16-8095-2016, 2016.

Ulbrich, I. M., Canagaratna, M. R., Zhang, Q., Worsnop, D. R. and Jimenez, J. L.: Interpretation of Organic Components from Positive Matrix Factorization of Aerosol Mass Spectrometric Data., Atmos. Chem. Phys., 9, 2891, doi:10.5194/acp-9-2891-2009, 2009.

Zhang, Q., Alfarra, M. R., Worsnop, D. R., James, D., Coe, H., Canagaratna, M. R. and Jimenez, J. L.: Deconvolution and Quantification of Hydrocarbon-like and Oxygenated Organic Aerosols Based on Aerosol Mass Spectrometry Deconvolution and Quantification of Hydrocarbon-like and Oxygenated Organic Aerosols Based on Aerosol Mass Spectrometry, Environ. Sci. Technol., 39(13), 4938–4952, doi:10.1021/es048568l, 2005.

Zhou, S., Collier, S., Jaffe, D. A., Briggs, N. L., Hee, J., Sedlacek III, A. J., Kleinman, L., Onasch, T. B. and Zhang, Q.: Regional influence of wildfires on aerosol chemistry in the western US and insights into atmospheric aging of biomass burning organic aerosol, Atmos. Chem. Phys., 17(3), 2477–2493, doi:10.5194/acp-17-2477-2017, 2017.

---

## Referee Comment (RC2) · Anonymous Referee #1 · 5 Dec 2017

Summary:

Aerosol composition measurements from an Aerodyne AMS and single-particle ATOFMS are presented for approximately 10 days at a rural ground site in Northern Michigan. Observations and HYSPLIT back trajectories suggest the site was influenced by a mixture of "background", wildfire, and urban sources. Secondary organic aerosol was the dominant aerosol species by mass regardless of source region, and the majority of particles throughout the campaign were identified as having originated from biomass burning. While the ubiquitous nature of biomass burning in the region is

interesting, the paper might benefit from additional analysis to overcome the shortfalls of having such a short time series. The paper is certainly appropriate for ACP and should be published after satisfying the following critiques.

Major Comments:

1. What metric is being used to separate the wildfire and "regional background" time periods? Figure 1 would suggest that ozone and PM2.5 mass/number concentrations are of similar magnitude between the two sources. Likewise, the relative compositional breakdown is identical in Figure 3. I believe it is a bit misleading to label one of these times as "regional background" when they both seem equally influenced by diffuse smoke. If the authors have used a quantitative metric for isolating these sources, please highlight this method in the revised text. The similarity in back trajectories does not seem sufficient from Figure S2. I suggest only presenting "urban" and "background" sources.

2. Ozone mixing ratios are quite low throughout the measurement period, as low as 10-20 ppb during background and wildfire periods. This seems anomalous for summertime North America. Please provide more context/validation for these measurements, especially with respect to your conclusion that the region was frequently influenced by transported smoke.

3. Your result of persistent biomass burning influence is somewhat in contrast to previous reports that biogenic emissions play a role in SOA production (and thus aerosol mass concentrations), e.g., Sheesley et al. (2004). Has something fundamentally changed in the transport pathways to the region, or can these differences be attributed to the VERY short duration of observations presented here?

4. The AMS and ATOFMS have very different sampling capabilities with respect to particle size. This is evident in the vastly different mass concentrations presented in Figure 3 and Figure 5. Can you comment on comparisons between the two measurements, in the context of measured size distributions shown in Figure S4. Do you think

the number-based prevalence of biomass burning is consistent with AMS results given that you are only assessing the tail of the mass distribution?

5. Rationale for this work is attributed to model under-prediction of OC in the Great Lakes region (i.e., in the abstract line-4), but the source of SOA is not entirely clear from the paper as written. The real question that I don't think is answered with this study is whether the precursor organic material contributing to the significant SOA mass coating biomass burning cores originated from the fires or from more-local biogenic sources encountered during transport. Can AMS (or ATOFMS) mass spectra help assess the SOA source? AMS source apportionment often utilizes the positive matrix factorization (PMF) technique to answer this sort of question and might provide very useful insight into SOA formation processes that could inform the model/measurement disconnect.

6. The buildup of urban emissions culminating in, presumably, a frontal passage at night on 7/22 is an interesting aspect of the sampling period. During this time, it seems that sulfate aerosol is potentially being produced during transport and, as you state, aerosol properties like hygroscopicity and CCN activity are likely significantly altered. Can the authors comment on the meteorological conditions conducive to this sulfate production, and whether situations like this might be an important moderator of aerosol chemistry in the region?

Minor Comments:

Page/Line: Comment

8/2: It is not clear at all from the N and M size distributions that this statement is true. Figure S4 shows the smallest dM/dlogD peak for urban cases, opposite of the conclusion in the text. Please revise and be more specific.

8/25: The presence of Na and Ca attributed to Great Lakes or seawater sources is interesting. I understand that the additional manuscript with likely cover these details, but it might be useful to state here whether ratios are consistent with sea-salt or con-

tinental dust. Further, soil dust is often lofted into the atmosphere in conjunction with biomass burning, is there any evidence for this during smoke transport? Whether this dust was coated with SOA or sulfates would also be a nice result, based on ATOFMS measurements.

10/3: Does the AMS mass spectrum support your hypothesis that these background particles are mostly SOA coated on a biomass burning core? m/z=60 has typically been used to quantify organic aerosols from biomass burning, can a lack of m/z=60 contribution be used to support the fact that most of the organics are not from combustion?

13/5: Given the source region for the biomass burning emissions, fires were more likely from agricultural burning. Were there any notable differences that could be highlighted in the mass spectrum or chemical properties of these particles compared to wildfires?

---

## Author Comment (AC1) · 1 Feb 2018

**Response to Reviewer #1**

**We thank Reviewer #1 for their helpful comments and suggestions. The original comments are provided below in gray, and our responses, with specific revisions, noted in bold font.**

Aerosol composition measurements from an Aerodyne AMS and single-particle ATOFMS are presented for approximately 10 days at a rural ground site in Northern Michigan. Observations and HYSPLIT back trajectories suggest the site was influenced by a mixture of "background", wildfire, and urban sources. Secondary organic aerosol was the dominant aerosol species by mass regardless of source region, and the majority of particles throughout the campaign were identified as having originated from biomass burning. While the ubiquitous nature of biomass burning in the region is interesting, the paper might benefit from additional analysis to overcome the shortfalls of having such a short time series. The paper is certainly appropriate for ACP and should be published after satisfying the following critiques.

Major Comments:

What metric is being used to separate the wildfire and "regional background" time periods? Figure 1 would suggest that ozone and PM2.5 mass/number concentrations are of similar magnitude between the two sources. Likewise, the relative compositional breakdown is identical in Figure 3. I believe it is a bit misleading to label one of these times as "regional background" when they both seem equally influenced by diffuse smoke. If the authors have used a quantitative metric for isolating these sources, please highlight this method in the revised text. The similarity in back trajectories does not seem sufficient from Figure S2. I suggest only presenting "urban" and "background" sources.

> **Wildfire and regional background periods were differentiated by the presence of smoke at the field site as determined by the NOAA HMS smoke maps, shown in Figure 4 and described on P11 L2-4. We clarified this during the regional background section (Section 3.2) on P9 L13-15, which now states "This remote background air mass period was differentiated from the wildfire influenced periods (Section 3.3) based on the lack of smoke impacting the site, as indicated by NOAA Smoke and Fire products (Figure 4)." While average ozone is similar, average $PM_{2.5}$ mass and number concentrations are significantly different between wildfire and regional background time periods as indicated by a t-test (P11 L 7-10). In addition, chemically-resolved ATOFMS mass concentrations (Figure 5) show lower biomass burning aerosol mass concentrations during the regional background period, compared to the rest of the study, when smoke influence was detected by the NOAA HMS smoke maps.**

Ozone mixing ratios are quite low throughout the measurement period, as low as 10-20 ppb during background and wildfire periods. This seems anomalous for summer- time North America. Please provide more context/validation for these measurements, especially with respect to your conclusion that the region was frequently influenced by transported smoke.

> **The range in ozone concentrations (9-63 ppb) are similar to previous measurements conducted at UMBS during background periods (~10-90 ppb) (Cooper et al., 2001); this statement was added to P10 L2-3. While ozone increased significantly with biomass burning influence during the first wildfire period, ozone levels were only slightly above background**

**during the second period, which also featured lower biomass burning aerosol concentrations (Figures 1 and 5). It should be noted that previous studies have shown that ozone does not always increase within wildfire plumes (P11 L17-19) (Jaffe and Wigder, 2012).**

Your result of persistent biomass burning influence is somewhat in contrast to previous reports that biogenic emissions play a role in SOA production (and thus aerosol mass concentrations), e.g., Sheesley et al. (2004). Has something fundamentally changed in the transport pathways to the region, or can these differences be attributed to the VERY short duration of observations presented here?

**Summer 2014 was unique in that it was one of the worst (largest) Canadian wildfire seasons in the past 25 years (P15 L17-19); therefore, increased influence from biomass burning would be expected. Wildfire intensity and frequency are increasing due to climate change (Gillett et al., 2004; Knorr et al., 2016; Liu et al., 2010; Veira et al., 2016). We added a statement noting this on P16 L 10-13, which states "The observed levels of biomass burning aerosol influence are attributed to the abnormally active Canadian wildfire season of 2014, compared to previous typical summers in northern Michigan with primarily biogenic SOA influence (Sheesley et al., 2004)."**

The AMS and ATOFMS have very different sampling capabilities with respect to particle size. This is evident in the vastly different mass concentrations presented in Figure 3 and Figure 5. Can you comment on comparisons between the two measurements, in the context of measured size distributions shown in Figure S4. Do you think the number-based prevalence of biomass burning is consistent with AMS results given that you are only assessing the tail of the mass distribution?

**The NOAA HMS smoke maps show satellite-based detection of smoke over the field site (Figure 4), and this is consistent with the abundance of biomass burning aerosols detected by the ATOFMS. We now indicate the AMS and ATOFMS size ranges in Figure S4, which shows the measured aerosol size distributions, to illustrate the overlap between the instruments and with the aerosol population. The mode diameter of biomass burning aerosols upon emission is typically between 100-180 nm (Reid et al., 2005); therefore, based on the full mass-based size distributions in Figure S4, it would be expected that the majority of the mass corresponds to biomass burning particles coated with SOA. Also, Figure 5 shows that biomass burning aerosols dominated the 0.5-2.0 μm aerosol mass, as measured by ATOFMS.**

Rationale for this work is attributed to model under-prediction of OC in the Great Lakes region (i.e., in the abstract line-4), but the source of SOA is not entirely clear from the paper as written. The real question that I don't think is answered with this study is whether the precursor organic material contributing to the significant SOA mass coating biomass burning cores originated from the fires or from more-local biogenic sources encountered during transport. Can AMS (or ATOFMS) mass spectra help assess the SOA source? AMS source apportionment often utilizes the positive matrix factorization (PMF) technique to answer this sort of question and might provide very useful insight into SOA formation processes that could inform the model/measurement disconnect.

**As stated in the response to Reviewer #1, while we agree that additional HR - AMS analysis would further support our findings, such analyses are unfortunately beyond the scope of what is feasible at this current time due to changes in personnel appointments following the field measurements. Therefore, we are unable to further apportion the SOA using the AMS data. However, we added discussion (P12 L17-19) of a previous SOA source apportionment study by Sheesley et al. (2004) who used molecular tracers, showing that the majority of summertime SOA in northern Michigan is from biogenic sources. Given the significant wildfire influence during our study, it is plausible that wildfire VOC precursors also contributed to SOA production.**

The buildup of urban emissions culminating in, presumably, a frontal passage at night on 7/22 is an interesting aspect of the sampling period. During this time, it seems that sulfate aerosol is potentially being produced during transport and, as you state, aerosol properties like hygroscopicity and CCN activity are likely significantly altered. Can the authors comment on the meteorological conditions conducive to this sulfate production, and whether situations like this might be an important moderator of aerosol chemistry in the region?

**We added the following sentences to P13 L12-14, "Stagnant air (wind speeds of ~ 2 m/s) led to the buildup of the urban-influenced PM, which peaked on July 22, as shown in Figure 1. The passing of a cold front, along with precipitation and a change in wind direction, led to a sudden decrease in PM concentrations late on July 22 (Figure 1)."**

Minor Comments

8/2: It is not clear at all from the N and M size distributions that this statement is true. Figure S4 shows the smallest dM/dlogD peak for urban cases, opposite of the conclusion in the text. Please revise and be more specific.

**Thank you for pointing this out. Upon further investigation we determined that the labels for the urban and wildfire periods were reversed and labeled incorrectly in this figure. Figure S4 (as well as Figure S3) are now labeled correctly and support the conclusions within the text.**

8/25: The presence of Na and Ca attributed to Great Lakes or seawater sources is interesting. I understand that the additional manuscript with likely cover these details, but it might be useful to state here whether ratios are consistent with sea-salt or continental dust. Further, soil dust is often lofted into the atmosphere in conjunction with biomass burning, is there any evidence for this during smoke transport? Whether this dust was coated with SOA or sulfates would also be a nice result, based on ATOFMS measurements.

**The ratios and other mass spectral ion peaks were not consistent with mineral dust, as stated on P8 L26-P9 L2.**

10/3: Does the AMS mass spectrum support your hypothesis that these background particles are mostly SOA coated on a biomass burning core? m/z=60 has typically been used to quantify organic aerosols from biomass burning, can a lack of m/z=60 contribution be used to support the fact that most of the organics are not from combustion?

**We now show a plot of AMS *m/z* 60 and 73 vs time as Figure S6, showing that biomass burning organic aerosol contributed to the OA measured and supporting the ATOFMS results in Figure 5.**

13/5: Given the source region for the biomass burning emissions, fires were more likely from agricultural burning. Were there any notable differences that could be highlighted in the mass spectrum or chemical properties of these particles compared to wildfires?

**Thank you for this suggestion. We now specify that the likely sources were an agricultural fire in Missouri and a forest fire in Alabama. As shown in Figure 6, a larger number fraction (58%) of the biomass burning particles detected during the urban air mass influence (agricultural burning influence) contained ammonium, compared to the Canadian wildfire periods (39%, by number). This is consistent with the AMS ammonium mass concentration being the highest of the study during this period and is attributed to agricultural ammonia emissions. This is described on P14 L12-14.**

[revised manuscript text omitted]

from vacuum aerodynamic diameter to mobility diameter) made by the AMS (blue) and ATOFMS (green) throughout the study are notated above the figure.

[Figure]

Figure S5. Ammonium balance calculated from predicted ammonium versus measured ammonium from the HR-AMS, following the method of Sueper (2010). A 1:1 line is showed in black for reference.

[Figure]

Figure S6. PM$_1$ non-refractory organic mass concentrations measured by HR-AMS (Figure 3), as well as specific tracers associated with biomass burning (*m/z* 60, $C_2H_4O_2^+$, and *m/z* 73, $C_3H_5O_2^+$) for comparison.

---

## Author Comment (AC2) · 1 Feb 2018

**Response to Reviewer #2**

**We thank Reviewer #2 for their helpful comments and suggestions. The original comments are provided below in gray, and our responses, with specific revisions, noted in bold font.**

Gunsch et al. present observations of aerosol concentration and composition at the University of Michigan Biological Station (UMBS) for July 2014. The authors use a combination of a high-resolution time-of-flight aerosol mass spectrometer, single particle aerosol time-of-flight mass spectrometer, and combination scanning mobility particle sizer spectrometer and aerodynamic particle size spectrometer to investigate the aerosol characteristics between 0.01 – 2.5 µm during this time period. The authors found four different air masses impacted the area during the time period: 2 air masses impacted by wildfires from Canada, 1 air mass impacted by cities south of UMBS, and 1 air mass from clean regions over Canada. The authors found an increase in particle number and mass, over the clean regime, for the air masses impacted by wildfires and cities; however, no matter where the air came from, it was always influenced by biomass burning. The paper provides important information about what influences the aerosol mass in a background, rural location, and is of value for Atmospheric Chemistry and Physics community; however, there are some concerns and some clarifications that need to be addressed first prior to publication.

Major Comments:

I'm wondering why the AMS data was not used more to help further validate the results from the single particle mass spectrometer, or to support some of the authors' hypotheses. For example, either PMF (Ulbrich et al., 2009), "poor-man's PMF" (Aiken et al., 2009; Zhang et al., 2005), or triangle plots of different fragments (Cubison et al., 2011; Hu et al., 2015; Ng et al., 2010) would further support the evidence of OA being strongly influenced by biomass burning, anthropogenic emissions, and biogenic emissions/chemistry (e.g., page 10, Lines 3 – 5 and page 11, lines 20 – 22). Without this support, speculations that biogenic emissions and chemistry leading to the very high O/C ratios observed is hard to interpret (page 11, lines 20 – 22). Other studies found high O/C ratios due to photochemical aging of the biomass burning emissions and aerosol (Liu et al., 2016; Zhou et al., 2017), which may have led to the high O/C instead of OA production from biogenic emissions. Also, cloud processing of the gases and OA may lead to high O/C ratios, along with the SO4 (Sullivan et al., 2016). A discussion either including these various processes or a discussion that argues why one process over the others leads to the high O/C ratios is needed.

> **We added a time series plot of AMS *m/z* 60 and 73 (levoglucosan markers) to Figure S6, showing that biomass burning organic aerosol contributed to the OA measured and supporting the ATOFMS results in Figure 5**. **These data are now referred to on P12 L11-14 and P14 L3-5.** **On page 12, we now clarify the discussion and summarize that the observed elevated O/C ratio is likely due to both photochemical and aqueous-phase oxidation of both biogenic and biomass burning VOC precursors. Additional references were added as suggested. While we agree that extensive HR - AMS analysis would further support our findings, such analyses are unfortunately beyond the scope of what is feasible at this current time due to changes in personnel appointments following the field measurements. Given this limitation, we instead focus here in**

**detailed ATOFMS analyses and use the HR - AMS to support our findings, which we feel is a best compromise approach.**

Throughout the Sections 3.2 – 3.4, in the comparisons of the different aerosol regimes, the aerosol number mode is mentioned. The biomass burning mode is the same as the background mode, and the urban mode is smaller than both the biomass burning and background mode; however, the authors discuss how chemistry and accumulation are occurring during transport of the biomass burning and urban air masses. A discussion about why the modes are similar (or smaller) while chemistry and accumulation is occurring is necessary for the readers to better relate these two possibly contradictory processes.

**In order to explain the smaller urban air mass mode, P13 L19-22 now reads: "The particle mode of 69 ± 29 nm was also the smallest of the study (Figure S3) due to contributions from combustion emissions, typically less than 50 nm (Seinfeld and Pandis, 2016), which likely grew to the observed sizes due to the condensation of secondary species during transport. A similar mode of 84 ± 18 nm was observed by VanReken et al. (2015) at UMBS during summer 2009 urban air mass influence." Based on the HYSPLIT backward air mass trajectories during urban air mass influence, the transport time from the nearest urban center was 24-36 h, which is less than the transport time from the Canadian wildfires (48 - 72 h), although it is difficult to compare expected particle growth during transport due to expected differences in precursor and oxidant species and concentrations.**

**To explain the similarities between the background and wildfire influence, P11 L10-12 now reads: "The particle number mode during wildfire influence was 80 ± 46 nm, similar to the background air mass period (mode of 82 ± 37 nm) (Figure S3)." As shown in Figure 3, the bulk submicron aerosol mass was similar between the background and wildfire influenced periods and dominated by oxygenated organics, showing that, while smoke influenced the site, the majority of the aerosol mass was secondary organic aerosol during both air mass influences. This is consistent with the significant transport time between the Canadian wildfires and the sampling location (48-72 h). While the ATOFMS identified the majority of >0.5 μm particles as having biomass burning cores, the majority of the aerosol mass on these particles was secondary organic aerosol (P11 L24-27).**

A more in - depth analysis of some of the aerosol characteristics would improve the results and the paper. For example, page 12, lines 6 – 11, the authors very briefly discuss an accumulation of SO 4 into the biomass burning particles; however, these particles are coming from wildfires in Canada. A discussion about where this SO 4 comes from would be beneficial, as recent studies have found a minor role for biomass burning emissions SO 4 and unclear for SO 2 emissions (Collier et al., 2016; Liu et al., 2016, 2017) .

**This is now addressed on P12 L 10-13: "$SO_2$ has been previously shown to be emitted from wildfires (e.g. Burling et al., 2010; Stockwell et al., 2014), and increases in particulate sulfate mass have been observed during wildfire plume aging through cloud processing (DeBell et al., 2004; Pratt et al., 2010)."**

As another example, the authors found little NO 3 in the air masses impacted by urban regions (page 14, lines 7 – 10 and Figure 3) , which is surprising with the NO x emissions and chemistry. What would have led to the extremely low amounts, especially since other downwind sites have observed enhanced NO 3 (Jimenez et al., 2009) ?

**During urban influence, the AMS measured the highest concentrations of nitrate throughout the study (Figure 3). The present study is most comparable to the New England USA (urban downwind) and Pinnacle Park NY (remote) sites within Jimenez et al. (2009); these sites both have similar percentages of nitrate by mass compared to our study. These points are now stated on P14 L14-16.**

Minor Comments
1) Page 1, line 27: " The field site was also influenced . . . " . It is not clear in the abstract what the influences are before this line.

**We deleted "also".**

2) Page 2, line 21: Change leading primary particles to leading to primary particles

**We made the suggested change.**

3) Page 2, line 22: Why is water included as a secondary species for aerosol?

For most particles (with sea spray aerosol as a notable exception), water vapor partitions from the gas-phase to the particle phase, making water a secondary species.

4) Page 4, line 3: Any reason why Slowik et al. (2011) is not included in this comparison of SOA in Ontario?

**This citation was added to P4 L4.**

5) Page 5, line 19: Please check the Jimenez and DeCarlo, 2017 citation. The website listed for this citation leads to Canadian Interagency Forest Fire Centre.

**This mistake was corrected.**

6) Page 11, line 1: superscript the minus sign

**This typo was corrected.**

7) Page 11, line 12: Please change limit to limiting

**This typo was corrected.**

[revised manuscript text omitted]

from vacuum aerodynamic diameter to mobility diameter) made by the AMS (blue) and ATOFMS (green) throughout the study are notated above the figure.

[Figure]

Figure S5. Ammonium balance calculated from predicted ammonium versus measured ammonium from the HR-AMS, following the method of Sueper (2010). A 1:1 line is showed in black for reference.

[Figure]

Figure S6. PM$_1$ non-refractory organic mass concentrations measured by HR-AMS (Figure 3), as well as specific tracers associated with biomass burning (*m/z* 60, C$_2$H$_4$O$_2^+$, and *m/z* 73, C$_3$H$_5$O$_2^+$) for comparison.